# Etiology of Anemia in Older Mexican Adults: The Role of Hepcidin, Vitamin A and Vitamin D

**DOI:** 10.3390/nu13113814

**Published:** 2021-10-27

**Authors:** Vanessa De la Cruz-Góngora, Aarón Salinas-Rodríguez, Mario Flores-Aldana, Salvador Villalpando

**Affiliations:** 1Center for Evaluation and Survey Research, Instituto Nacional de Salud Pública (National Institute of Public Health), Cuernavaca 62100, Mexico; vcruz@insp.mx; 2Center for Nutrition and Health Research, Instituto Nacional de Salud Pública (National Institute of Public Health), Cuernavaca 62100, Mexico; mario.flores@insp.mx (M.F.-A.); svillalp@insp.mx (S.V.)

**Keywords:** anemia, vitamin A, vitamin D, hepcidin, inflammation, older adults

## Abstract

Anemia in older adults is a growing public health issue in Mexico; however, its etiology remains largely unknown. Vitamin A deficiency (VAD) and vitamin D deficiency (VDD) have been implicated in the development of anemia, though by different mechanisms. The aim of this study is to analyze the etiology of anemia and anemia-related factors in older Mexican adults. This is a cross-sectional study of 803 older adults from the southern region of Mexico in 2015. The anemia etiologies analyzed were chronic kidney disease (CKD), nutritional deficiencies (ND), anemia of inflammation (AI), anemia of multiple causes (AMC) and unexplained anemia (UEA). VAD was considered to be s-retinol ≤ 20 μg/dL, and VDD if 25(OH)D < 50 nmol/L. IL-6 and hepcidin were also measured. Multinomial regression models were generated and adjusted for confounders. Anemia was present in 35.7% of OA, independent of sex. UEA, CKD, AI and ND were confirmed in 45%, 29.3%, 14.6% and 7% of older adults with anemia, respectively. Hepcidin and log IL-6 were associated with AI (*p* < 0.05) and CKD (*p* < 0.001). VAD was associated with AI (*p* < 0.001), and VDD with ND and AMC (*p* < 0.05). Log-IL6 was associated with UEA (*p* < 0.001). In conclusion, anemia in older adults has an inflammatory component. VAD was associated to AI and VDD with ND and AMC.

## 1. Introduction

Anemia in older adults is a serious health problem that affects quality of life and predicts short-term survival [1]. Although national survey data have revealed a significant and increasing prevalence of anemia in older adults, no study has characterized the magnitude of its main causes in the Mexican population [2].

Significant variations have been documented in the magnitude of each known cause of anemia in older adults, due to different study criteria, methods and populations [3]. Nonetheless, one third of this condition has been attributed to chronic inflammation and CKD, one third to nutritional deficiencies, and a similar proportion (≈30%) remains of unknown etiology [4].

Chronic comorbidities become more frequent with age and may represent a leading risk factor for anemia in the elderly population. Anemia of inflammation (AI) refers to immunologically based anemia mediated by inflammatory cytokines, which control hepcidin expression, thereby blocking iron exportation and resulting in poor iron availability for cells [5].

Identifying immunomodulatory metabolites implicated in the development of AI may contribute to prevention and treatment options. Vitamin A (VA) and vitamin D (VD) have a crucial role in modulating the immune response and regulating a broader range of gene expression [6]. Both vitamins have been implicated in the development of anemia, suggesting that they play a role in the iron mobilization metabolic pathway [7,8]. Previous studies have explored the association between VD deficiency (VDD) with AI [9,10]; however, evidence of an association between VA deficiency (VAD) and anemia in older adults is scarce. Studies in rodents have suggested a possible link between VA status and hepcidin, the mediator of AI [11,12].

Evidence of VA and VD status as contributing to anemia in older adults is scarce. Causal and risk factors of anemia may contribute differently across populations, particularly in developing countries. Therefore, the aim of this study is to identify possible causes of anemia as well as to analyze anemia-related factors in older Mexican adults. First, we explored the magnitude of each identified etiology of anemia. Then, we explored whether or not hepcidin levels were higher in AI or lower in iron deficiency anemia (IDA). Finally, we examined the potential association between VAD and VDD status, with AI.

## 2. Materials and Methods

### 2.1. Population and Data Collection

A cross-sectional study in older adults aged 60 and older from the southern region of Mexico was carried out from August to September 2015. A stratified, multistage cluster sample design was used. The final analysis included 803 individuals aged 60 and older with complete serum data from an original sample of 829. Demographic, socioeconomic, health and nutritional information were collected, using ad hoc questionnaires.

Participants were interviewed at home, and each provided written informed consent. The study was approved by the Research, Ethics, and Biosecurity Committees of the National Institute of Public Health, Mexico (INSP, for its Spanish acronym). The study protocol was registered at ClinicalTrials.gov as NCT04820465.

### 2.2. Laboratory Analysis

Fasting venous blood samples were drawn and centrifuged in situ. Serum was separated and stored in coded cryovials, then stored at −20 °C in liquid nitrogen until delivery to a central laboratory in Cuernavaca, Mexico.

Capillary hemoglobin was measured, using a portable photometer (Hemocue). C-reactive protein (CRP mg/L), homocysteine (μmol/L), B12 (pg/mL), ferritin (ng/mL), VitD-25OH (nmol/L) and folate (ng/mL) were measured through chemiluminescent immunoassay, and creatinine (mg/dL) through colorimetry, using commercials kits (Abbott Diagnostics) and an ArchitectCI8200 autoanalyzer. Serum retinol was determined in an HPLC HP1110 LCDAD (Agilent Technology, Waldbronn, Germany), using NovaPack columns C18 4 um 3.9 × 150 mm with a flow of 1.5 mL/min of mobile phase methanol, after extraction with 99% ethanol. Serum iron was measured by an atomic absorption analyzer (Beckton Dickinson), and soluble transferrin receptor (sTfR) by a commercial immunoassay kit, using recombinant human sTfR as the standard (Quantikine IVD sTfR ELISA kit; R&D Systems Inc., Minneapolis, MN, USA). Hepcidin was measured, using a commercial immunoassay kit (My Biosource ELISA) with a detection range from 4.69 ng/mL to 300 ng/mL. Alpha glycoprotein 1-acid (AGP), erythropoietin (EPO) and interleukin 6 (IL-6) were measured by immunoassay ELISA, using commercials kits (R&D Systems Inc., Minneapolis, MN, USA).

Biochemical analyses were performed at the Centro Médico Nacional Siglo XXI and in the nutrition laboratory of the INSP.

### 2.3. Definition of Variables

Indigenous background was defined as any indigenous language spoken in the household. A household wealth index was constructed using principal component analysis, then divided into tertiles, with the uppermost tertile representing the highest socioeconomic status. Anthropometric data (weight and height) were collected, using validated and standardized methods. Body mass index (BMI) was estimated and grouped into two categories: normal (18.5–24.9 kg/m^2^), and overweight/obese (≥25 kg/m^2^) [13]. Chronic comorbidities, including hypertension, diabetes, dyslipidemia, myocardial infarction, angina pectoris, heart disease, cirrhosis, arthritis, stroke, chronic lung disease, osteoporosis, and cancer were identified through self-report of any previous diagnosis by a physician. Functional status was based on the Katz index for activities of daily living (ADL) and Lawton scale for instrumental ADLs (IADLs) [14,15]. Pharmaceutical drug consumption was classified as the use of non-steroidal anti-inflammatory drugs (NSAID) or steroidal anti-inflammatory drugs (SAID). Frailty phenotype was determined, according to a modified version of the model proposed by Fried et al. [16]. Vitamin D deficiency was defined as 25(OH)D < 50 nmol/L [17], and vitamin A deficiency as s-retinol ≤ 20 μg/dL [18].

### 2.4. Etiology of Anemia

Anemia and severity of anemia was defined, according to the World Health Organization (WHO) guidelines; anemia was defined as Hb < 12 g/dL in women and <13 g/dL in men [19]. Anemia related to chronic kidney disease (CKD) was considered as having an estimated glomerular filtration rate (eGFR) of <60 mL/min/1.73 m^2^ [20], or a prior physician diagnosis of renal disease. ID was defined as s-ferritin < 15 ng/mL after correcting for inflammation [21], or as sTfR > 28 nmol/L. If no evidence of CKD or ID was detected, AI was defined as (1) the presence of cirrhosis, cancer or low serum iron (<60 μg/dL) in the absence of ID, (2) CRP > 5 mg/L and AGP > 1 g/L, or 3) s-ferritin ≥ 350 ng/mL [22]. Since ID may coexist with chronic inflammation, renal dysfunction or cancer, older adults with ID with previous disease diagnosis were reclassified either as having CKD or AI etiology. Vitamin B12 deficiency (B12D) was defined as <−0.5 SD, considering homocysteine and folate [23]. Folate deficiency (FD) was defined as s-folate < 4 ng/mL [24]. Anemia of multiple causes (AMC) was defined as the coexistence of nutritional deficiencies (ND) with AI and CKD. If older adults with anemia could not be classified into any of these categories, they were considered to have unexplained anemia (UEA).

### 2.5. Statistical Analysis

Sample characteristics are displayed as mean and SD or proportion, and for variables with biased distributions as medians and interquartile ranges. Bivariate associations for categorical variables of each identified cause of anemia compared with the non-anemia group were tested, using a Pearson chi-square analysis. For continuous variables, a quantile regression model was used. In both cases, adjustments were made, using the Bonferroni correction. A Pearson’s correlation between hepcidin and biomarkers of inflammation was performed, using log-transformed variables.

A multinomial logistic regression model clustered by state was used to identify associations between hepcidin concentrations, VAD, VDD and other factors related to each identified cause of anemia. Models were adjusted by sex, age, ethnicity, socioeconomic status, IL-6 level, frailty, BMI, NSAID consumption and type 2 diabetes.

Statistical significance was set at α = 0.05. Analyses were performed in STATA SE V15 (College Station, TX, USA).

## 3. Results

Descriptive characteristics of older adults are shown in Table 1; 10.9% of older adults had VB12D, 5% ID, 9% VDD, and 2.5% VAD. Anemia affected 35.7% of older adults independent of sex (*p* = 0.957). Older adults with anemia had higher prevalence of functional disability, inflammation, frailty, diabetes, arthritis, cancer, VB12D, VAD and VDD than those without anemia (Table 1).

Table 2 shows the distribution of anemia severity and possible causal factors. Mild anemia was prevalent (85.9%). IDA accounted for just 1.1% of older adults with anemia. Nutritional deficiencies accounted for 7% of total anemia, mainly due to VB12D. No FD was detected within this population. CKD accounted for 29.3% of all causes of anemia. AI was present in 14.6% of older adults with anemia, and 45% of anemia was unexplained.

Participant characteristics and serum biomarker data by presence and type of anemia are reported in Table 3 and Table 4. In the non-anemia group, participants were significantly younger than those with any etiology of anemia (Table 3). The prevalence of VAD and CRP > 10 mg/L was significantly higher in AI, as compared to participants without anemia (*p* < 0.05). Levels of inflammatory markers (hepcidin, CRP, AGP and IL-6) were significantly higher in AI as compared to participants without anemia (*p* < 0.05), and IL-6 was higher in CKD; IL-6 and CRP were also significantly higher in AMC.

CRP, hepcidin and AGP levels were unchanged in UEA as compared to participants without anemia (*p* > 0.05) (Table 4). Prevalence of functional disability was significantly higher in AI and CKD, and frailty in AI, as compared to participants without anemia (*p* < 0.05). No significant differences were found by sex, NSAID or other pharmaceutical drug consumption, or VDD between groups (Table 3 and Table 4).

In the univariate analysis, Hb was negatively and significantly correlated with log-normalized hepcidin levels across the sample (r = −0.09, *p* = 0.0083)—in participants with anemia (r = −0.18, *p* = 0.0019) and without anemia (r = −0.07, *p* = 0.07)—and was positively correlated with log-normalized ferritin (r = 0.10, *p* = 0.0022). A negative correlation was revealed in UEA between log EPO and Hb (r = −0.25, *p* = 0.0029).

In the multinomial logistic regression models, factors associated with higher odds of anemia varied in comparison with the non-anemia group (Table 5). In the adjusted model, hepcidin, log of IL-6, age, NSAID consumption and diabetes were conditions associated with CKD anemia (*p* < 0.05). Older adults with VDD were associated with ND anemia and AMC (*p* < 0.05). Hepcidin, VAD, IL-6, higher age, indigenous background and NSAID consumption were associated with higher odds of AI (*p* < 0.05). VDD, IL-6 and frailty were associated with AMC (*p* < 0.05). IL-6, pre-frailty, age, female sex, indigenous background and NSAID consumption were associated to UEA (*p* < 0.05) (Table 5).

## 4. Discussion

To our knowledge, this study is the first to explore the main etiologies of anemia in ambulatory older Mexican adults and the association of selected micronutrient deficiencies with anemia. In our population, anemia in older Mexican adults has an inflammatory component, characterized by higher hepcidin and IL-6 levels in AI and CKD, while in UEA an association was revealed with IL-6. VAD was associated with AI; however, VDD was not. Although VDD was associated with overall anemia, nutritional anemia and multiple-cause anemia, the null association between VDD and AI contrasted with our initial hypothesis.

In order to better describe the underlying etiology of anemia in older adults, an in-depth literature review was performed [25,26,27,28,29,30,31,32,33]. Nevertheless, a high proportion of the anemia detected could not be classified through any of the known criteria, and therefore, remained of unexplained etiology. Among the identified causes, CKD was predominant in our study population, while nutritional anemias, particularly IDA, showed the lowest contribution.

In our study, chronic inflammation played an important role for anemia, with hepcidin apparently being an important mediator. As hypothesized, higher hepcidin levels were found in subjects with AI, and hepcidin was associated with CKD. AI has no standardized definition by clinical parameters, and is usually associated with an underlying condition (cancer, infections, CKD or autoimmune disease) [34].

Among the anemia etiologies analyzed in our study, CKD contributed the most, in contrast to previous studies where the contribution of CKD was lower [35]. This may be explained by variability in the criteria used to define CKD, the diagnostic thresholds applied or population of interest. In our study, diabetes was a strong predictor of CKD anemia. Diabetes is a serious health problem in Mexico, affecting 25.5% of adults over 70 years old [36]; it is the second leading cause of death in older adults, and alongside CKD, is among the top three causes of reduced disability-adjusted life years in Mexico [37]. Strategies for diabetes prevention may reduce the burden of comorbidities associated with anemia, including renal damage and, therefore, CKD anemia.

The levels of hepcidin associated with CKD in our study may be due in part to the characteristic inflammatory processes of the disease, through a decrease in renal clearance function, or by the uremic toxins produced by damaged nephron cells, which stimulate hepcidin synthesis [38]. This result is consistent with previous studies, where hepcidin levels were shown to be higher in AI and in CKD anemia [28,29], as compared to participants without anemia. Although AI and CKD were consistently associated with higher hepcidin and IL-6 levels than in participants without anemia, we observed no correlation between hepcidin and inflammation biomarkers (CRP, AGP and IL-6).

Nutritional deficiencies, particularly ID, show a low contribution to anemia in the study population, reflecting previous studies [9,39]. Contrary to our hypothesis, hepcidin levels were no different among participants with IDA as compared to those without anemia, and in the IDA group, no correlation was observed between ferritin and hepcidin concentrations. These findings contrast with previous studies [26,28,29], where hepcidin levels were significantly lower among older adults with IDA, as diagnosed by lower ferritin levels. Although ferritin was the indicator used to define ID in our study, its concentration increases with age and inflammation. Distinguishing IDA from AI is challenging, due to the variability of s-ferritin values (<30 ng/mL), which may be lower in both conditions, contributing to false-negative test results, even after adjustment for inflammation [21,40]. This adjustment did not change the ID prevalence estimated (delta 0.25 pp, data not shown).

No study participants showed folate deficiency. The folate status was measured in serum, which may not have captured the long-term folate status as red blood cells would have; however, since the introduction of folic acid fortification in staple foods, FD is no longer considered a public health priority [41]. Vitamin B12 was the predominant nutritional deficiency in this study, which is a similar result to previous studies [35].

A high proportion of anemia detected in study participants could not be explained by the identified etiologies. UEA prevalence was similar to that reported in other studies, and since this was an exclusion criterium, the heterogeneity of etiologies not measured was high (rare hematopoietic disorders, myelodysplastic syndromes, blunted erythropoietin response, etc.) [33,42]. In the adjusted model, the log of IL-6 predicted UEA. Artz et al. [43] reported a significant correlation between levels of neopterin and Hb (r = −0.459, *p* = 0.048), suggesting an underlying low-grade proinflammatory profile in the pathophysiology of UEA in older adults. Frailty in older adults is characterized by an inflammatory profile driven mainly by IL-6 [44]. In our study, pre-frailty and NSAID consumption were associated to UEA, after considering confounders. This may suggest a role of low-grade pro-inflammatory in the development of anemia of unknown cause.

### 4.1. Role of VAD

Studies on VAD and anemia in older adults are scarce since in developed countries, VAD is no longer considered a health priority [45,46]. In our study, VAD prevalence was higher in older adults with anemia than those without anemia, and associated to AI and to a lesser extent with CKD. VA has an important role in hematopoiesis and iron metabolism [8] through favoring the differentiation and proliferation of red blood cells, as well as in modulating EPO expression [47] and strengthening immunity to infections [6,48].

Experimental studies in rodents have highlighted a possible relation between VA and hepcidin through enhancing the expression of HAMP in VAD [11,12]. In a previous study using the same sample [49], the analysis of the association between VA and hepcidin showed that the VAD status was associated with a two-fold positive association to hepcidin concentrations, suggesting that the relationship of VAD to anemia may manifest itself through inflammatory pathways. Residual confusion due to inflammation may explain the association between VAD and AI.

### 4.2. Role of VDD

The exact mechanism by which VD affects the development of anemia is unknown, but some studies have proposed that VD stimulates erythroid precursors and downregulates pro-inflammatory cytokine expression [50,51]. VD was previously associated with AI as a direct suppressor of HAMP mRNA transcription [51,52]; however, as in previous studies, we found no association between VDD and hepcidin [49] or VDD and AI [53,54]. Some authors have suggested that 1,25(OH) rather than 25(OH) is a greater predictor of anemia risk [9,55]. Our results showed an association between 25(OH)D and anemia, but contrary to our expectations, none were found between VDD and AI or CKD; VDD was, however, associated with nutritional deficiencies and AMC. Further studies are needed to explore whether the active form of VD, 1,25(OH)2D, may represent a better predictor. VDD and frailty were both associated with AMC. One potential explanation for this is reverse causality since anemia may be a consequence and not a cause of multiple comorbidities. Older adults may experience greater frailty and poor health status, which leads to less time outdoors and less sun exposure, and consequently, higher VDD prevalence.

### 4.3. Study Strengths and Limitations

Anemia is a multifactorial condition. Defining a single etiology for anemia with a cross-sectional design is challenging since multiple conditions are present in the same person; reverse causality may be an alternative explanation in any association observed. In our study, we selected diagnostic criteria according to those most frequently identified and documented in epidemiological studies to define the etiology of anemia; nevertheless, caution should be used when comparing these results to those of other populations, given variations in the criteria used to define etiological factors as well as the population characteristics. The prevalence of inflammation was high in our sample as compared with other studies exploring anemia causes in other older adult populations.

In addition, thresholds and ranges for most biomarkers tested (i.e., ferritin, retinol, eGFR, Hb and hepcidin) must be defined and validated for older adults.

The causes of anemia in older adults in Latin America have not been comprehensively analyzed. This study is a first approach toward identifying the diverse causes of anemia in older adults from Mexico, where the prevalence of anemia remains high. Further longitudinal studies are needed to explore the underlying causes of anemia, specifically the role of VA and VD in anemia onset, in order to identify opportunities for treating or preventing anemia of known etiology.

## 5. Conclusions

In summary, anemia was highly prevalent in the study population of older adults. CKD and inflammation were the main causes of anemia of known etiology as characterized by higher levels of hepcidin and IL-6. A high proportion of older adult participants had unexplained anemia. Vitamin A deficiency, and not vitamin D deficiency, was associated with anemia of inflammation, while vitamin D deficiency was associated with nutritional deficiencies anemia and anemia of multiple causes.

## Figures and Tables

**Table 1 nutrients-13-03814-t001:** Descriptive characteristics of older adult study population by anemia condition.

Characteristic	Total (%)	Non-Anemia Group n = 516 (%)	Anemia Group n = 291 (%)	*p* Value *
Sex (female)	60.9	60.5	61.3	0.821
Age group, years				
60–69	49.2	54.3	40.1	
70–79	34.5	33.9	35.5	
80+	16.3	11.8	24.4	<0.001
Indigenous background	33.7	30.4	39.7	0.008
Household wealth index				
Tertile 1	33.5	29.9	40.1	
Tertile 2	34.8	36.1	32.3	
Tertile 3	31.7	34	27.7	0.014
Body mass index, range				
Underweight	0.9	0.2	2.2	
Normal	21.8	17.2	30.1	
Overweight	38	40.1	34.1	
Obese	39.4	42.5	33.7	<0.001
CRP > 5 mg/L	33.7	31	38.7	0.029
AGP > 1 g/L	7.6	5.2	11.8	<0.001
IL-6 > 10 pg/dL	10.7	7.6	16.4	<0.001
Functional disability				
ADL	29.5	24.2	39	<0.001
IADL	42.3	36.6	52.6	<0.001
Frailty				
Not frail	39.7	44.6	31.1	
Pre-frail	47.2	45.3	50.5	
Frail	13.1	10.1	18.5	<0.001
Comorbidities as previously diagnosed by physician				
Diabetes	30.4	28.1	34.5	0.066
Hypertension	49.9	49	51.6	0.508
Dyslipidemia	35.2	34.1	37.3	0.397
Renal disease	12.3	12.2	12.5	0.911
Arthritis	20.4	18.4	24	0.068
Cirrhosis	2.4	1.7	3.5	0.146
Cancer	4.2	2.7	7.0	0.006
Pharmaceutical drug consumption (yes)	74.0	71.5	78.4	0.036
NSAID consumption (yes)	15.8	14	19.2	0.056
Current smoker (yes)	24.8	26.7	21.3	0.089
Serum micronutrient deficiency				
B12 deficiency	9.0	8.1	10.5	0.303
Iron deficiency	5.1	4.5	6.3	0.315
Vitamin A deficiency	3.4	1.7	6.3	0.002
Vitamin A deficiency (adj.) **	2.6	1.5	4.5	0.008
Vitamin D deficiency	9.7	7.75	13.2	0.018

* χ^2^ test. Abbreviations: ADL: activities of daily living; IALD: instrumental activities of daily living; NSAID: non-steroidal anti-inflammatory drugs; CRP: C-reactive protein; AGP: Alpha glycoprotein 1 acid. Vitamin B12 deficiency was defined as <−0.5 SD, considering Fedosov’s equation; iron deficiency as s-ferritin <15 ng/mL after correcting for inflammation, or if sTfR > 28 nmol/L; vitamin A deficiency (VAD) as s-retinol ≤ 20 μg/dL; vitamin D deficiency if 25(OH)D < 50 nmol/L. ** Adjusted VAD by inflammation (CRP and AGP).

**Table 2 nutrients-13-03814-t002:** Severity of anemia and contributions of possible causal factors in older adults.

Variable	Frequency (%)
Anemia	35.7
Severity of anemia	
Mild	85.9
Moderate	13.8
Severe	0.3
Nutritional deficiencies	7.0
ID	1.1
VB12D	5.6
ID + VB12D	1.1
Chronic kidney disease	29.3
Inflammation	14.6
Multiple causes	3.1
Unexplained	45.0

Abbreviations: ID = iron deficiency, VB12D = vitamin B12 deficiency.

**Table 3 nutrients-13-03814-t003:** Descriptive characteristics of older adult study population, by etiology of anemia.

	Non-Anemia Group	Chronic Kidney Disease	Iron Deficiency	Inflammation	B12-Deficiency	Multiples Causes	Unknown
Variable/(n sample)	(516)	(84)	(6)	(42)	(16)	(9)	(130)
Sex (female)	60.5	59.5	50	52.4	56.3	66.7	66.2
Indigenous background	30.4	35.7	0	47.6	43.8	22.2	42.3
Age group, years							
60–69	54.3	34.5	50	28.6	31.3	44.4	47.7
70–79	33.9	34.5	50	40.5	37.5	44.4	33.1
80+	11.8	31	0	31	31.3	11.1	19.2
Household wealth index							
Tertile 1	29.9	43.9	16.7	39	37.5	66.7	37.5
Tertile 2	36.1	25.6	33.3	34.1	31.3	22.2	36.7
Tertile 3	34	30.5	50	26.8	31.3	11.1	25.8
Body mass index, range							
Normal	17.4	24.4	40	46.2	37.5	22.2	32.8
Overweight/obese	82.6	75.6	60	53.8 *	62.5	77.8	67.2
Comorbidities as previously diagnosed by physician							
Type 2 diabetes	28.1	48.8 *	0	26.2	37.5	33.3	29.2
Hypertension	49	64.3	66.7	35.7	50	77.8	46.2
Dyslipidemia	34.1	46.4	50	16.7	25	22.2	40
Cancer	2.7	9.5 *	0	23.8 *	0	22.2	0
Cirrhosis	1.7	3.6	0	14.3 *	0	11.1	0
Renal disease	12.2	39.3 *	0	0	0	33.3	0
Arthritis	18.4	25	50	28.6	6.3	11.1	23.8
Alcohol consumption	31.4	19	16.7	26.2	25	22.2	22.3
Current smoker	26.7	20.2	33.3	19	25	44.4	20
Functional disability							
ALD	24.2	44 *	16.7	57.1 *	25	66.7	30.8
IALD	36.6	57.1 *	16.7	66.7 *	50	88.9	44.6
Not frail	44.6	25	66.7	33.3	25	11.1	34.6
Pre-frail	45.3	52.4	0	38.1	56.3	33.3	56.2
Frail	10.1	22.6 *	33.3	28.6	18.8	55.6 *	9.2
Pharmaceutical drug consumption (yes)	71.5	94	66.7	61.9	75	77.8	74.6
NSAID consumption (yes)	14	19	50	14.3	18.8	22.2	19.2
SAID consumption (yes)	2.7	2.4	0	2.4	6.3	11.1	0.8
Anemia severity							
Mild	-	77.1	83.3	88.1	81.3	55.6	93.8
Moderate	-	22.9	16.7	11.9	12.5	44.4	6.2
Severe	-	0	0	0	6.3	0	0

Abbreviations: activities of daily living (ADL), instrumental activities of daily living (IADL), non-steroidal anti-inflammatory drug (NSAID), steroidal anti-inflammatory drug (SAID), C-reactive protein (CRP), Alpha glycoprotein 1 acid (AGP). * Statistically different from the non-anemia group, *p* < 0.05 after adjustment by Bonferroni correction.

**Table 4 nutrients-13-03814-t004:** Serum biomarkers of older adult study population, by cause of anemia.

	Non-Anemia Group	Chronic Kidney Disease	Iron Deficiency	Inflammation	Vitamin B12 Deficiency	Multiple Cause	Unknown Cause
Variable/(n sample)	(516)	(84)	(6)	(42)	(16)	(9)	(130)
Vitamin D, range							
25(OH)D ≥ 75 nmol/L	50.4	41.7	66.7	52.4	31.3	44.4	47.7
25(OH)D 50–74 nmol/L	41.9	42.9	0	38.1	56.3	11.1	42.3
25(OH)D < 50 nmol/L	7.8	15.5	33.3	9.5	12.5	44.4 *	10
Retinol ≤ 20 μg/dL	1.7	6	0	19 *	6.3	0	3.1
Retinol ≤ 20 μg/dL (adj.) ^1^	1.6	4.6	0	11.4 *	5.1	0	2.9
Vitamin B12 deficiency	8.1	6	50 *	0	100	66.7 *	0
Iron deficiency	4.5	7.1	100	7.1	0	33.3 *	0
Ferritin < 15 ng/mL	1.6	0	33.3 *	4.8	0	0	0
Serum iron < 60 μg/dL	4.1	16.7 *	50 *	54.8 *	18.8	44.4 *	0
Ferritin ≥ 350 ng/mL	7	11.9	0	11.9	0	33.3	3.8
CRP, range							
0–3 mg/L	49.6	44	100	26.2	50	33.3	53.8
3–10 mg/L	36.8	36.9	0	28.6	31.3	33.3	40
>10 mg/L	13.6	19	0	45.2 *	18.8	33.3	6.2
AGP (>1 g/L)	5.2	16.7 *	0	28.6 *	12.5	44.4 *	1.5
IL6 (>10 pg/dL)	7.6	17.9 *	0	42.9 *	12.5	33.3	6.9
Mean of biochemical biomarkers ^a^							
25(OH)D (ng/mL)	30.1 (25.3–36.5)	27.9 (23.6–33.4)	33.9 (18.2–43.3)	30.9 (23.7–42.4)	28.3 (25.7–38.1)	26.7 (17.9–38.1)	29.6 (24.2–34.2)
Retinol (nmol/L)	45 (36.3–55.6)	52.8 (42.6–73) *	39.4 (26.3–49.3)	37.8 (28.5–44.8)	42.9 (29.2–47.7)	31.3 (28.5–33.1)	45.3 (37.6–52.3)
Hepcidin (mg/dL)	13.1 (5.3–27.3)	15.2 (5.2–35.6)	16.7 (10.7–32.8)	23 (5.9–43.1) *	12.3 (7.2–30.6)	18.8 (13.2–42.7)	9.2 (4.2–22)
CRP (mg/L)	3 (1.6–6.5)	3.6 (1.2–8.8)	1.2 (0.8–1.6)	7.7 (2.6–37.7) *	3.1 (0.9–7.6)	6.4 (2.1–15.5)	2.8 (1.6–6.4)
AGP (g/L)	0.5 (0.4–0.7)	0.6 (0.5–0.8)	0.6 (0.5–0.7)	0.7 (0.4–1.1) *	0.6 (0.4–0.8)	0.9 (0.8–1.2) *	0.6 (0.4–0.7)
IL6 (pg/mL)	2.5 (1.4–4.5)	4.5 (2.6–8) *	2.3 (0.2–3.7)	5.7 (2.3–21.4) *	3.8 (1.7–5.2)	8.8 (7.8–18.4) *	2.6 (1.6–4.6)
Iron (μg/dL)	100.5 (83.9–123.6)	79.8 (65.3–109.7) *	72 (53.4–88.9)	58.8 (48–86.1) *	88.3 (63.2–111.7)	88.7 (55.6–108)	99.2 (79.3–129)
Erythropoietin (miu/mL)	10 (7.8–13.4)	11.1 (8.2–16.1)	18.6 (15.1–28.5)	12.3 (8.3–17.5)	12.9 (9.5–16.8)	10.4 (7.1–14.3)	10.5 (7.4–13.4)
Ferritin (ng/mL)	120.8 (75.7–198.6)	122.1 (77.7–203.1)	54.5 (13.1–84.6)	121.7 (44.5–184.2)	96.3 (72.7–129.6)	244.8 (145.3–355.6) *	118.8 (74.7–171.5)
Pharmaceutical drugs consumed, number ^b^	3.3 ± 2.3	4.3 ± 2.7 *	4.8 ± 3.6	4 ± 3.2	3.6 ± 2.1	4.4 ± 3.2	3.7 ± 2.4
Chronic comorbidities, number ^b^	2 ± 1.6	3.1 ± 1.7 *	2.5 ± 0.8	2.1 ± 2	1.8 ± 1	3.1 ± 1.7	2 ± 1.7
Hemoglobin (g/dL) ^b^	13.7 ± 1.2	10.9 ± 1.1	11 ± 1.3	11.2 ± 1.1	11 ± 1.7	10.2 ± 1.2	11.3 ± 0.9

^a^ Median (interquartile range). ^b^ Mean ± SD. * Statistically different from the non-anemia group, *p* < 0.05 adjusted by the Bonferroni correction. ^1^ Retinol levels adjusted for inflammation (CRP and AGP). Abbreviations: C-reactive protein (CRP), Alpha glycoprotein 1 acid (AGP), Interleukine-6 (IL6).

**Table 5 nutrients-13-03814-t005:** Multinomial regression model of variables associated with each cause of anemia.

	Chronic Kidney Disease	Nutritional	Inflammation	Multiple Cause	Unknown Cause
Variables *	OR (CI95%)	OR (CI95%)	OR (CI95%)	OR (CI95%)	OR (CI95%)
Hepcidin, ng/mL	1.01 (1.01, 1.01)	1.01 (1.00, 1.02)	1.01 (1.00, 1.02)	1.01 (0.99, 1.03)	0.99 (0.97, 1.01)
VA deficiency	2.26 (0.95, 5.41)	2.44 (0.02, 300)	3.20 (2.31, 4.44)	0 (0, 0)	1.35 (0.31, 5.81)
VD deficiency	1.40 (0.47, 4.20)	1.80 (1.04, 3.1)	0.86 (0.07, 9.96)	5.76 (4.78, 6.93)	1.11 (0.81, 1.53)
Log IL-6	1.29 (1.17, 1.43)	0.97 (0.72, 1.32)	1.91 (1.30, 2.82)	3.68 (1.45, 9.35)	1.08 (1.04, 1.11)
Frailty					
Pre-frail	1.33 (0.89, 1.97)	0.95 (0.53, 1.68)	0.81 (0.09, 7.59)	1.02 (0.83, 1.25)	1.39 (1.07, 1.82)
Frail	1.51 (0.26, 8.64)	1.51 (0.78, 2.93)	1.15 (0.03, 47.32)	5.76 (5.39, 6.17)	0.83 (0.78, 0.88)
Overweight/obese	0.99 (0.81, 1.22)	0.36 (0.08, 1.68)	0.44 (0.15, 1.29)	1.94 (0.55, 6.87)	0.42 (0.25, 0.72)
Age	1.08 (1.06, 1.09)	1.02 (0.93, 1.13)	1.05 (1.00, 1.10)	1.03 (0.9, 1.18)	1.02 (1.01, 1.02)
Sex	0.82 (0.73, 0.91)	0.66 (0.08, 5.22)	0.88 (0.51, 1.53)	0.55 (0.11, 2.63)	1.26 (1.05, 1.51)
Indigenous background	0.83 (0.6, 1.14)	0.98 (0.71, 1.36)	1.48 (1.13, 1.95)	0.48 (0.16, 1.46)	1.47 (1.22, 1.77)
Household wealth index, tertile					
2	0.47 (0.22, 0.99)	0.86 (0.64, 1.14)	1.26 (0.55, 2.9)	0.15 (0.08, 0.26)	0.93 (0.57, 1.51)
3	0.72 (0.58, 0.89)	1.05 (0.19, 5.76)	1.41 (1.23, 1.62)	0.08 (0, 2.43)	0.82 (0.44, 1.53)
NSAID consumption	1.85 (1.06, 3.24)	3.24 (0.72, 14.49)	1.19 (1.16, 1.22)	1.38 (0.37, 5.17)	1.6 (1.17, 2.2)
Type 2 diabetes	2.87 (1.16, 7.11)	1.16 (0.65, 2.09)	0.72 (0.16, 3.24)	0.99 (0.63, 1.56)	1.1 (0.32, 3.82)
Intercept	0 (0, 0)	0.01 (0, 381.11)	0 (0, 0)	0 (0, 17.35)	0.11 (0.09, 0.12)

Abbreviations: vitamin A (VA), vitamin D (VD), non-steroidal anti-inflammatory drugs (NSAID). * For each categorical variable, the reference values are, respectively, no VA or VD deficiency, non-frailty, male sex, tertile 1 of HWI, normal BMI, non-indigenous, non-consumption of NSAIDs and without diabetes previously diagnosed by a physician.

## Data Availability

The data that support the findings of this study are available from the corresponding author, A.S.-R., upon reasonable request.

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
