# Peer review of "Etiology of Anemia in Older Mexican Adults: The Role of Hepcidin, Vitamin A and Vitamin D"

_nutrients, 2021, doi:10.3390/nu13113814_

Round 1
Reviewer 1 Report
This study on the etiology/epidemiology of anemia in older adults in Mexico involves a reasonably large population, was carefully performed, had appropriate statistical analysis, and is set in the context of an effective review of the relevant literature. Novel features of the manuscript include its focus on the older adult population of Mexico, which has some unique ethnographic features; and the discussion of vitamin A deficiency, which tends to be under-emphasized in the Western hemisphere. The collection of data on household wealth relevant to the accumulated information is also important.
The paper is clearly written. My suggested modifications are minor and related purely to enhancing the presentation.
- The use of the abbreviation “OA” for “older adults” is potentially confusing. It is a commonly used abbreviation for “osteoarthritis”, a common condition in older adults and one which has a complex relationship to inflammatory anemias. I would suggest not using an abbreviation and just referring to “older adults” as “older adults”.
- The authors refer to their patients as “anemics” or “non-anemics”. While this is clear terminology and not pejorative, most journals and books prefer to use either “anemic patients” or “patients with anemia” (and an equivalent expression for individuals who are not anemic) or even “anemic persons”
Author Response
We really appreciate the reviewers’ comments. We have incorporated all the suggested changes in the new version of the manuscript Etiology of anemia in older Mexican adults: the role of hepcidin, vitamin A and vitamin D.
- The use of the abbreviation “OA” for “older adults” is potentially confusing. It is a commonly used abbreviation for “osteoarthritis”, a common condition in older adults and one which has a complex relationship to inflammatory anemias. I would suggest not using an abbreviation and just referring to “older adults” as “older adults”.
The abbreviation “OA” was removed from the entire document, and it was referred as “older adults”. Thank you for the comment.
- The authors refer to their patients as “anemics” or “non-anemics”. While this is clear terminology and not pejorative, most journals and books prefer to use either “anemic patients” or “patients with anemia” (and an equivalent expression for individuals who are not anemic) or even “anemic persons”
The suggested change was applied in the manuscript, and it was referred as: “older adults with anemia/without anemia”.

Reviewer 2 Report
This manuscript is generally well written. It is difficult to identify specific etiology of anemia. However, the authors clearly defined the definition for this study, and described it. While sample size is small and not very representative, a study from Latin America has merit.
Several comments.
The terminology and abbreviations needs some attention. For example, CKD and renal disease are same, but they are mixed in Tables. OA for older adults seems to be not necessary abbreviation.
Table 1. Table 3.
It is confusing because CKD is defined by physician diagnosis in these Tables, and the author noted " Anemia related to chronic kidney disease 109 (CKD) was considered as estimated glomerular filtration rate (eGFR)<60 mL/min/1.73m2 110 [20], or a prior physician diagnosis of kidney disease" in methods section. GFR and CKD defind by GFR need to be noted.
Table 2: What does 35.7 means on the right side of anemia severity?
Author Response
We really appreciate the reviewers’ comments. We have incorporated all the suggested changes in the new version of the manuscript Etiology of anemia in older Mexican adults: the role of hepcidin, vitamin A and vitamin D.
The terminology and abbreviations needs some attention. For example, CKD and renal disease are same, but they are mixed in Tables. OA for older adults seems to be not necessary abbreviation.
The terminology used for “renal disease” and “CKD” was corrected and clarified in the document. The abbreviation “OA” was removed, and it was referred as “older adults”.
Table 1. Table 3.
It is confusing because CKD is defined by physician diagnosis in these Tables, and the author noted "Anemia related to chronic kidney disease 109 (CKD) was considered as estimated glomerular filtration rate (eGFR)<60 mL/min/1.73m2 110 [20], or a prior physician diagnosis of kidney disease" in methods section. GFR and CKD defind by GFR need to be noted.
Thank you for noting this. The terms were corrected and clarified in the document to avoid confusion. In Table 1 and Table 3, the variable “renal disease” is referenced as only previously diagnosed by a physician, while the variable “CKD” as a cause of anemia is defined either by previous physician diagnosis or by estimated glomerular filtration rate (eGFR)<60 mL/min/1.73m2, as noted in line 121-123.
Table 2: What does 35.7 means on the right side of anemia severity?
- There was a mistake on reporting this. The 35.7 correspond to the total frequency of anemia. This has been corrected.
